# Comparative Analyses of the *Exopalaemon carinicauda* Gut Bacterial Community and Digestive and Immune Enzyme Activity during a 24-Hour Cycle

**DOI:** 10.3390/microorganisms10112258

**Published:** 2022-11-14

**Authors:** Shumin Xie, Runyao Liu, Huiling Zhang, Fei Yu, Tingting Shi, Jiawei Zhu, Xinlei Zhou, Binlun Yan, Huan Gao, Panpan Wang, Chaofan Xing

**Affiliations:** 1Jiangsu Key Laboratory of Marine Bioresources and Environment, Jiangsu Ocean University, Lianyungang 222005, China; 2Jiangsu Key Laboratory of Marine Biotechnology, Jiangsu Ocean University, Lianyungang 222005, China; 3Co-Innovation Center of Jiangsu Marine Bio-Industry Technology, Jiangsu Ocean University, Lianyungang 222005, China; 4Lianyungang Marine and Fishery Development Promotion Center, Lianyungang 222044, China; 5Marine Resource Development Institute of Jiangsu (Lianyungang), Lianyungang 222005, China; 6The Jiangsu Provincial Infrastructure for Conservation and Utilization of Agricultural Germplasm, Nanjing 210014, China

**Keywords:** intestinal bacterial community, *Exopalaemon carinicauda*, circadian rhythm, digestive enzyme activity, immune enzyme activity

## Abstract

The change in life activities throughout a cycle of approximately 24 h is called the circadian rhythm. Circadian rhythm has an important impact on biological metabolism, digestion, immunity, and other physiological activities, but the circadian rhythm of crustaceans has rarely been studied. In this study, the activity of digestive enzymes (α-amylase, trypsin, and lipase) and immune enzymes (superoxide dismutase, lysozyme, and catalase), as well as the circadian rhythm of the intestinal bacterial community of *Exopalaemon carinicauda*, were studied. The results showed that the digestive and immune enzyme activities of *E. carinicauda* changed significantly (*p* < 0.05) at four time points throughout the day by one-way ANOVA analysis, with the highest value at 24:00 and the lowest value at 12:00. The highest values of alpha diversity and richness were observed in the 24:00 samples, which were significantly higher than those in the other groups (*p* < 0.05). The principal coordinate analysis (PCoA) results obviously showed that the samples from the same sampling time had higher similarity in the bacterial community structure. *Candidatus hepatoplasma* had the highest abundance among the intestinal microorganisms at 24:00, and *Marinomonas* had the highest abundance at 12:00. This study contributed to the understanding of digestive enzyme activity, immune enzyme activity, and the circadian rhythm of the intestinal bacterial community structure in *E. carinicauda*. It will play an important role in optimizing feeding times and improving digestion and nutrient utilization for *E. carinicauda*. The results of this study provide a basis for further study on the physiological mechanism of diurnal variation of intestinal flora in crustaceans.

## 1. Introduction

The 24-hour cyclical change in environmental conditions, such as light and temperature caused by the Earth’s rotation, affects biological and physiological activities in what is known as the circadian rhythm [1]. The circadian rhythm of aquatic animals manifests as a series of changes in metabolism, immunity, and other activities due to changes in the nervous and endocrine systems [2]. The activities of aquatic biological immune defense [3,4,5], and metabolism [6] are affected by variation over the course of a day. Biological clocks interact with gut microbes to maintain metabolic homeostasis by regulating cell and organ functions [7]. Any imbalance between the state of intestinal flora and circadian rhythms can lead to imbalances in related systems and can affect the balance of metabolic functions [8]. 

As a powerful environmental cue, the timing of food intake has the potential to destroy or restore the synchrony of circadian rhythms in metabolism [9]. Feeding, immunity, and other physiological activities have been used to research the circadian rhythm in the sharpsnout seabream (*Diplodus puntazzo*) [10], the whiteleg shrimp (*Litopenaeus vannamei*) [11], *Haliotis discus hannai* [12], the flathead gray mullet (*Mugil cephalus*) [13], the Nile tilapia (*Oreochromis niloticus*) [5], etc. In a self-feeding system, *L. vannamei* displays nocturnal feeding and locomotor rhythms, with activity peaking in darkness and decreasing at the end of the dark phase [11]. Digestive and immune enzyme activity are important indicators for studying the digestive absorption and immune defense of aquatic animals [14,15]. The relationship between digestive enzyme activity and feeding mechanisms has been studied in *O. niloticus* [16], *Eriocheir sinensis* [17], *Patinopecten yessoensis* [18], and *L. vannamei* [19]. Meanwhile, the relationship between immune enzyme activity and the immune defense mechanism has been studied in *Takifugu rubripes* [16], *L. vannamei* [20], and the zebrafish *Danio rerio* [21]. 

Intestinal microorganisms in an organism are interdependent and mutually restricted and maintain the homeostasis of the internal environment [22]. As an indivisible ‘organ’ of the organism, the gut bacterial community has a great influence on the physiological activity of the host, such as immunity and growth, and its composition is related to many factors such as heredity, nutrition, and the environment [23,24,25,26]. Some mammalian studies have found that the gut bacterial community changes regularly during the daily cycle, and both the circadian clock and the gut microbiota influence each other in a reciprocal fashion [7,8,27,28]. The gut bacterial community exerts an important influence on the growth development, digestion-absorption, and immune defense of shrimp [25,28,29,30]. Yu et al. found that the alpha diversity and richness of the gut bacterial community from *E. Sinensis* were highest in samples collected at 18:00, and daily variation may affect the innate immune response of *E. Sinensis* [31].

*Exopalaemon carinicauda* is one of the major economically important shrimp species raised in aquaculture systems in China and belongs to the Palaemonidae family of the subphylum Crustacea [32,33]. As an economically important shrimp breeding species, *E. carinicauda* exhibits nocturnal hunting behavior. In this study, healthy *E. carinicauda* were selected and sampled at four time points in a day to compare the digestive and immune enzyme activities, and intestinal bacterial community structure of *E. carinicauda* at different time points. Studying the daily changes in intestinal microbial composition and determining the optimal feeding time will provide ideas for improving aquaculture production efficiency. 

## 2. Materials and Methods

### 2.1. Ethics Statement and Sample Collection

The experimental protocols and procedures for animal husbandry and handling were performed according to the Animal Care Committee of Jiangsu Ocean University in the present study. In total, 100 healthy *E. carinicauda* individuals (weight: 1.43 ± 0.21 g) were obtained from aquaculture pond of Jiaxin Fishery Development Company Ltd. in Lianyungang (Jiangsu, China). All individuals were transported to the Jiangsu Key Laboratory of Marine Biotechnology of Jiangsu Ocean University and acclimated for two weeks in environmentally controlled tanks (salinity 25, temperature 25, natural light period). During the two-week acclimation phase, the shrimp were temporarily fed twice daily (07:00, 18:00) with artificial compound feed (Haida, Guangzhou, China) at a daily feeding rate of 5% body weight. Feeding was stopped 24 h before the experiments. During the experimental period, 20 individuals were randomly collected at four time points, at intervals of 6 h (12:00, 18:00, 24:00, and 6:00). The shrimp were euthanized with the anesthetic (alcohol–eugenol = 10:1). The intestinal tract, hepatopancreas, and stomach tissues of individuals were sampled and immediately preserved in liquid nitrogen. The stomach, hepatopancreas, and intestinal tissues were used to study the digestive enzyme activity, immune enzyme activity, and intestinal flora structure, respectively.

### 2.2. Enzyme Activity Determination

Enzyme extracts were prepared to determine the enzyme activities of α-amylase (AMS), trypsin (TRS), lipase (LIP), superoxide dismutase (SOD), lysozyme (LZM), and catalase (CAT). The stomach (four individuals per sample) was used to determine the activity of three digestive enzymes, and the hepatopancreas (four individuals per sample) was used to determine the activity of three immune enzymes. The total protein content was determined by Coomassie brilliant blue method. Weighed samples were mixed with 9 times the volume of normal saline (0.9%), which was homogenized for 20 s at 4 °C and 20 Hz/s using an automatic grinding machine (Tissuelyser-24, Shanghai, China). Centrifuge the sample at 4 °C and 1500× *g* for 10 min using a refrigerated centrifuge (MIKRO 220R, Hettich, Germany). Enzyme extracts were kept at −80 °C until they were analyzed within 48 h of extraction. Total protein content and all enzymatic activity analyses were conducted using commercial assay kits (Jiancheng Biological Engineering Institute, Nanjing, China). An AMS activity unit was 10 mg of starch hydrolyzed per milligram of protein in the tissue at 37 °C for 30 min and recorded at OD_660nm_. The unit of LIP activity was the consumption of 1 μmol of substrate per gram of protein in the tissue at 37 °C for 1 min and recorded at OD_420nm_. At pH 8.0 and 37 °C, the TRS contained in each gram of protein changed the absorbance by 0.003 per minute as an enzyme activity unit and was recorded at OD_235nm_. The activity of LZM was the amount of enzyme in the reaction system that led to the increase in light transmittance due to bacterial lysis under the conditions of 37 °C for two minutes and was recorded at T_530nm_. In the reaction system, when the inhibition rate of SOD reached 50%, the corresponding enzyme amount was one SOD activity unit and was recorded at OD_450nm_. Each milligram of tissue protein decomposed at 1 μmol of H_2_O_2_ per second as a CAT activity unit and was recorded at OD_405nm_. Data were presented as the means ± standard deviation, and one-way analysis of variance (ANOVA) was used to determine the significance of enzymatic activities.

### 2.3. Extraction of Genomic DNA and Amplicon Generation

Total genomic DNA was extracted from samples using CTAB/SDS. Using agarose gels (1%), the DNA concentrations and purity were monitored. Sterile water was used to dilute DNA to 1 ng/μL according to its concentration. 338F and 806R primers were used to amplify distinct 16S rRNA genes (V3-V4). The PCR system contained 15 μL Phusion^®^ High-Fidelity PCR Master Mix, 0.2 μM each primer, and 10ng target DNA. The PCR process was as follows, the first temperature reached 98 °C, began denaturation, and the time was 1 min. Then, 30 cycles were performed at a temperature of 98 °C for 10 s, a temperature of 50 °C for 30 s, and a temperature of 72 °C for 30 s. Finally, extension at 72 °C for 5 min. An agarose gel electrophoresis method was used to detect PCR products, which were recovered using a Qiagen Gel Extraction Kit (Qiagen, Hilden, Germany).

### 2.4. Library Preparation and Sequencing

The gut tissue of the fasting shrimp is nearly transparent, which is not easy to obtain smoothly. Twenty individuals were randomly collected at four time points. We successfully obtained intestinal tissue from 18 individuals at each sampling site, which was evenly divided into three groups. The NEBNext^®^ UltraTM IIDNA Library Prep Kit (Cat No. E7645) was used to generate 12 sequencing libraries according to the manufacturer’s instructions. We evaluated the quality of the libraries using a Qubit@ 2.0 Fluorometer (ThermoFisher Scientific, Waltham, MA, USA) and an Agilent 2100 Bioanalyzer (Agilent Technologies, California, CA, USA). With paired-end reads of 250 bp, these libraries were sequenced on the Illumina NovaSeq platform.

### 2.5. Bioinformatics and Statistical Analysis

Paired-end reads were assigned to the sample according to their unique barcode and were truncated by cutting off the barcode and primer sequences. The paired-end reads were merged using FLASH (Version 1.2.11, http://ccb.jhu.edu/software/FLASH/, accessed on 5 March 2022), and the splicing sequence was used to obtain the raw tags. Quality filtering of the raw tags was performed using fastp (Version 0.20.0) software to obtain high-quality clean tags. The clean tags were compared with the reference database (silva V. 138 database https://www.arb-silva.de/ (accessed on 5 March 2022) for 16S, unite 8.2 database https://unite.ut.ee/ for ITS, accessed on 5 March 2022) using Vsearch (Version 2.15.0) to detect chimera sequences, and then the chimera sequences were removed to obtain the effective tags. Denoising was performed on the effective tags with the DADA2 or deblur module in QIIME2 software (version QIIME2-202006) to obtain initial ASVs (amplicon sequence variants), and then ASVs with abundances lower than 5 were filtered out. Multiple sequence alignment was performed using QIIME2 software. The absolute abundance of ASVs was normalized using a standard sequence number corresponding to the sample with the fewest sequences.

Alpha diversity was calculated based on indices in QIIME2, including observed species, Chao1, Shannon, and Simpson. The Kruskal–Wallis test was used to analyze whether there were significant differences in species diversity between groups. The observed species index indicates how many species have been observed directly. Chao1 is the total number of species present in the community sample. The higher the Chao1 index is, the more low-abundance species there are in the community. The Shannon index is the total number of classifications and their proportions in the samples. The Simpson index shows the diversity and evenness of species distribution in the community. Weighted and unweighted distance calculations in QIIME2 were used to determine β diversity, evaluate the complexity of community composition, and compare the differences between samples.

Principal coordinate analysis (PCoA) was performed to obtain principal coordinates and visualize differences in samples in the complex multidimensional data. A matrix of weighted or unweighted UniFrac distances among the sample data was transformed into a new set of orthogonal axes, where the maximum variation factor was demonstrated by the first principal coordinate, the second maximum variation factor was demonstrated by the second principal coordinate, and so on. The three-dimensional PCoA results were displayed using the QIIME2 package, while the two-dimensional PCoA results were displayed using the ade4 package and ggplot2 package in R software (Version 2.15.3). To study the significance of the differences in community structure between groups, the adonis and anosim functions in QIIME2 software were used to perform the analysis. To determine the significantly different species at each taxonomic level (phylum, class, order, family, genus, species), R software (Version 3.5.3) was used to perform MetaStat and T-tests. LEfSe software (Version 1.0) was used to perform LEfSe analysis (LDA score threshold: 4) to determine the relevant biomarkers.

Furthermore, to study the functions of the microbiota communities in the samples and to determine the different functions of the communities at the different sample times, PICRUSt2 software (Version 2.1.2-b) was used for functional annotation analysis.

## 3. Results

### 3.1. The Circadian Rhythm of Digestive Enzyme Activity

The activity of α-amylase, lipase, and trypsin showed time-dependent changes (Figure 1). The activity of α-amylase was the lowest at 12:00 and gradually increased to its highest value at 24:00 (4.46 U·mgprot^−1^). The activity decreased at 6:00 but was higher than that at 18:00. The activity of AMS at 24:00 was significantly different from that at other times (*p* < 0.05, Figure 1A). The lipase activity was the lowest at 12:00 and gradually increased to the highest value at 24:00 (2.17U·mprot^−1^). The lipase activity decreased at 6:00 but was higher than that at 18:00 (Figure 1A). The trypsin activity was the lowest at 12:00 and gradually increased to the highest value at 24:00 (2212.12 U·mgprot^−1^) with time. The trypsin activity decreased at 6:00 but was still higher than that at 18:00. The TRS activity at 24:00 was significantly different from that at other sample times (*p* < 0.05, Figure 1A).

### 3.2. The Circadian Rhythm of Immune Enzyme Activity

The activities of lysozyme, total superoxide dismutase, and catalase are shown in Figure 1. The lysozyme activity was the lowest at 12:00 and gradually increased to the highest value at 24:00 (254.93 U·mgprot^−1^) with time and was significantly different from that of the other three sample times (*p* < 0.05, Figure 1B). The activity of total superoxide dismutase was the lowest at 12:00 and gradually increased to the highest value at 24:00 (22.02 U·mgprot^−1^) with time. There were significant differences with other sample times (*p* < 0.05, Figure 1B). The activity of catalase was the lowest at 12:00 and gradually increased to the highest value at 24:00 (13.93 U·mgprot^−1^) with time. The activity decreased at 6:00 but was still higher than that at 18:00. The activity of CAT at 24:00 was significantly different from that at other sample times (*p* < 0.05, Figure 1B).

### 3.3. Amplicon Sequence Variants

The original sequence of gut samples was filtered and denoised to obtain the amplicon sequence variants (ASVs). We analyzed the common and unique ASVs among the different sample times and presented the results as a Venn diagram. We screened 8682 ASVs, of which 2355, 3395, 3820, and 3362 ASVs were detected in the 12:00, 18:00, 24:00, and 06:00 samples (Figure 2). There were 66 ASVs that only appeared in the 12:00 and 18:00 samples, 90 ASVs that only appeared in the 12:00 and 06:00 samples, 165 ASVs that only appeared in the 24:00 and 18:00 samples, 83 ASVs that only appeared in the 06:00 and 18:00 samples, 115 ASVs that only appeared in the 24:00 and 06:00 samples, 82 ASVs that only appeared in the 24:00 and 12:00 samples. All groups shared 289 ASVs, which belonged to 10 phyla, 22 classes, 50 orders, 83 families, and 121 genera.

### 3.4. Alpha Diversity

Alpha diversity is an indicator of the richness and diversity of microbial communities in samples. Alpha diversity was used to analyze the microbial community diversity at the four time points (Figure 3). According to the Chao1, observed Shannon and Simpson indices, the change rule of the four time points was basically consistent. The microbial community richness and diversity of the 12:00 sample were significantly lower than those of the other three sample times (*p* < 0.05). The Chao1 and observed index of the 24:00 sample was significantly higher than that of the 12:00 sample (*p* < 0.01). The value of the Shannon index at 24:00 was significantly higher than that at 18:00 (*p* < 0.05), and the value at 18:00 was lower than that at 06:00. The community diversity and species evenness of the sample were the highest at 24:00.

### 3.5. Relative Abundance of the Gut Bacterial Community

Based on the species annotation results at different classification levels, a relative abundance histogram of species was generated, which shows an intuitive representation of the species composition and proportion of samples at different classification levels and at different time points. The relative abundances of various intestinal bacteria varied over the course of 24 h. Ten of the most abundant phyla were Proteobacteria, Firmicutes, Bacteroidota, Actinobacteriota, Patescibacteria, Campilobacterota, Desulfobacterota, Cyanobacteria, Bdellovibrionota, and Acidobacteriota (Figure 4A). Ten of the most abundant genera were *Colwellia*, *Leucothrix*, *Photobacterium*, *Sulfitobacter*, *Flavobacterium*, *Pseudomonas*, *Candidatus-hepatoplasma*, *Ralstonia*, *Candidatus hepatincola*, and *Marinomonas* (Figure 4B). The most abundant species were *Leucothrix mucor* and *Neptunomonas concharum*, collected at 24:00 and 06:00, respectively (Figure 4C).

### 3.6. Beta Diversity

Beta diversity is represented by the comparative analysis of microbial community composition in different samples. Using a series of eigenvalues and eigenvectors, principal coordinate analysis (PCoA) extracts the most significant elements and structures from multidimensional data. The PCoA analysis is carried out using the weighted and unweighted unifrac distance, and the main coordinate combination with the largest contribution rate is chosen for illustration. As shown in Figure 5, the PCoA results obviously showed that the samples from the same sampling time had higher similarity in the bacterial community structure. On the contrary, the similarity of bacterial communities among groups was lower. The smaller the distance between samples, the more similar the species’ composition is. The two-dimensional principal coordinate analysis of the gut bacterial community at the four sample times shows that the relative aggregation between the 12:00, 24:00, and 06:00 samples is more similar in the composition of bacteria within the aggregate group and more dispersed between the 18:00 sample and the other three samples.

### 3.7. Species Abundance

Based on the species annotation and abundance information of the samples at the phylum and genus levels, the top 35 phyla and genera were selected. According to the abundance information in each sample, clustering was performed from the two levels of species and samples, and a heatmap was drawn to facilitate the discovery of the concentration of species in each sample. The results indicated that both the phylum- and genus-level abundance of the bacterial communities varied significantly during the daily cycle. As shown in the phylum-level abundance cluster diagram (Figure 6A), Dependentiae, Thermoplasmatota, and Patescibacteria had high abundances only in the 12:00 sample; *Nitrospirota*, *Planctomycetota,* and *Euryarchaeota* had high abundances only in the 18:00 sample; *Latescibacterota*, *Methylomirabilota,* and *Actinobacteriota* had high abundances only in the 24:00 sample; *Nanoarchaeota*, *Crenarchaeota,* and *Armatimonadota* had high abundances only in the 06:00 sample. As shown in the genus-level abundance cluster diagram (Figure 6B), *Psychrobacter*, *Cupriavidus,* and *Pseudomonas* had high abundances only in the 12:00 sample; *Enterococcus*, *Albimonas,* and *Leucothrix* had high abundances only in the 18:00 sample; *Corynebacterium*, *Vibrio*, and *Colwellia* had high abundances only in the 24:00 sample; *Neptunomonas*, *Muribaculaceae*, and *Photobacterium* had high abundances only in the 06:00 sample. 

### 3.8. Gene Function of the Gut Bacterial Community

Based on the gene information of the ASV tree and ASV in the Greengenes database, a Phylogenetic Investigation of Communities by Reconstruction of Unobserved States 2 (PICRUSt2) was used to infer the gene function spectrum of the common ancestors for the community. Moreover, the gene function spectrum of other unknown species in the Greengenes database was deduced. Construction of the gene function prediction spectrum of archaeal and bacterial domains was carried out. Finally, the composition of the sequenced flora was mapped to the database to predict the metabolic function of the flora. As shown in the relative abundances of gene functions (Figure 7), the relative abundance of the ABC-type glycerol-3-phosphate transport system, pimeloyl-ACP methyl ester carboxylesterase, ABC-type sugar transport system, and other metabolic functions in the 12:00 sample was relatively large. The relative abundances of the Na^+^-driven multidrug efflux pump, permease of the drug/metabolite transporter (DMT) superfamily, nucleoside-diphosphate-sugar epimerase, and other metabolic functions in the 06:00 sample were relatively large. The relative abundances of the GGDEF domain, diguanylate cyclase (c-di-GMP synthetase), methyl-accepting chemotaxis protein, superfamily II DNA and RNA helicase, and other metabolic functions in the 24:00 sample were relatively large.

## 4. Discussion

The circadian rhythm is an endogenous timing system that regulates the function of cells and organs and synchronizes physiological and external cues to establish metabolic homeostasis [7]. The circadian rhythm is a regular metabolic cycle, and organisms regulate physiological activities such as digestion and immunity through the perception of the external environment [8]. The study of the circadian rhythms of organisms is beneficial to exploring the laws of life phenomena and establishing scientific breeding models.

In aquatic animals, the sharpsnout seabream *Diplodus puntazzo* has shown circadian rhythmicity in motor activity, feeding, and plasma melatonin under light conditions [10]. Flathead gray mullet *Mugil cephalus* somatic cell growth showed circadian rhythmicity under different feeding patterns [13]. In this study, we found that the activities of the digestive and immune enzymes showed circadian rhythmicity and were the highest at 24:00 and the lowest at 12:00. We speculate that this remarkable circadian rhythmicity in digestive and immune enzyme activities is regulated by the circadian clock. Some physiological activities, such as biological feeding, are more active at night and may be associated with avoiding predators. In tambaqui (*Colossoma macropomum*), Fortes et al. found a typical pattern for nocturnal locomotor activity (91%), and lower percentages of Gram-positive 6 × 10^3^, 24% and Gram-negative cocci (approximately 38%) were observed during the same periods compared with the diurnal period [34]. The *L. vannamei* and *Haliotis discus hannai* show circadian rhythmicity in sports activities under light/dark cycle conditions [11,12]. *Sinonovacula constricta* has shown circadian rhythmicity in digestive enzymes under light/dark cycle conditions [35]. Innate immune defenses exhibit circadian rhythmicity in Nile tilapia (*Oreochromis niloticus*) [5].

The gut bacterial community is closely related to the digestive metabolism of organisms. Gut bacteria have their daily rhythms in composition, intestinal niche positioning, and function [36]. The composition and functional structure of the intestinal microbial community are closely related to time and are both regulated by the host and able to affect the biological clock of the host [37]. The results of α-diversity and β-diversity analyses showed that there were significant differences in the abundance and diversity of intestinal bacteria at the four time points, and the trends were the same. The Chao1, Shannon, and Simpson indices showed that the richness and diversity of intestinal bacteria were the highest at 24:00, the richness and diversity of intestinal bacteria were relatively low at 18:00 and 06:00, and the richness and diversity of the intestinal bacterial community at 12:00 were significantly lower than other groups. Compared to the 06:00 group, the 18:00 group’s Shannon index was lower, while its Simpson index was higher. This may be because although both the Simpson index and the Shannon index can comprehensively evaluate the richness and evenness of community species composition, the Shannon index has the closest relationship with richness, while the Simpson index has a more distant relationship with richness. In this study, the Simpson index was more sensitive to enriched species, while the Shannon index was more sensitive to sparse species [38].

The relative abundances of the ten most abundant intestinal microorganisms at the four time points were compared at the phylum, genus, and species levels. The results showed that the intestinal bacterial community structure was different at different time points. The most abundant phyla are Proteobacteria, Firmicutes, and Bacteroidota. The most abundant genera are *Marinomonas*, *Ralstonia*, *Candidatus hepatoplasma*, and *Candidatus hepatincola*. The cluster diagram of species abundance shows that there are significant differences in the aggregation of species at different times. Studies have found that gut microbiota show different structural compositions at different time points in the day [39,40]. Studies have reported the relative abundance of Bacteroidetes, Firmicutes, and Proteobacteria [41]. Previous studies have detected high abundances of *Pseudomonas*, *Flavobacteria*, *Photobacterium*, and *Colwellia* in shrimp gut microbiota [42,43,44]. Studies have confirmed that *Photobacterium* and *Colwellia* can produce eicosapentaenoic acid (EPA) and docosahexaenoic acid (DHA), which belong to omega-3 polyunsaturated fatty acids [45]. *Neptunomonas concharum*, one of the bacteria with high relative abundance, is a kind of high polymer polysaccharide degrading bacteria that can produce poly-3-hydroxybutyrate (PHB) [46]. *Bifidobacterium thermophilus* is an important intestinal beneficial bacteria that plays an important role in nutrition, immune enhancement, and improving gastrointestinal function [47]. The intestinal microbiota analysis showed that dark treatment-induced alterations in intestinal bacterial abundances in *L. vannamei*, such as decreased (*p* < 0.05) relative abundance of *Formosa*, *Demequina*, *Lutimonas,* and increased (*p* < 0.05) relative abundance of *Ruegeria*, *Vibrio*, *Actibacter*, and *Roseovarius* at the genus level [48].

In this study, the function of intestinal flora was predicted, and it was found that the metabolic functions with large relative abundances were different at different time points. At 24:00, the abundance of DNA binding and transcription-related pathways was large. Studies have shown that the gut microbiota produces microbial metabolites that support host metabolism by decomposing nutrients that cannot be digested by the host [8]. The microbiota functional analysis demonstrated that the dark treatment group of *L. vannamei* mainly increased the susceptibility of pathogens, decreased nutrition metabolism, and influenced circadian rhythm [48]. The circadian clock optimizes the physiological environment for organisms by affecting the timing of the metabolic processes. The interaction between the host and microorganism is affected by feeding [39].

## 5. Conclusions

In this study, the circadian rhythm of digestive and immune enzyme activity, as well as the intestinal bacterial community of *E. carinicauda* were studied. The results showed that the activities of digestive and immune enzymes in *E. carinicauda* changed significantly at four time points throughout the day. The activities of three digestive enzymes (AMS, LIP, and TRS) and three immune enzymes (SOD, LZM, and CAT) all reached the highest value at 24:00, and the lowest value at 12:00. The structure, species diversity, relative abundance, and function of gut microbiota were varied at different periods. The intestinal microbial diversity was highest at 24:00 and lowest at 12:00. The most abundant phyla were Proteobacteria, Firmicutes, and Bacteroidota. The most abundant genera were *Marinomonas*, *Ralstonia*, *Candidatus hepatincola*, and *Candidatus hepatoplasma*. This study has some limitations, especially the influence of external environmental factors on circadian rhythm. The relationship between the composition of intestinal flora and digestive enzyme activity is unclear. Therefore, in future studies, it will be of great value to further study the external factors affecting the circadian rhythm of *E. carinicauda* and how the intestinal bacterial community affects its digestive function. These studies can further determine the optimal feeding time and improve digestion and nutrient utilization for *E. carinicauda*. In summary, our results improved the understanding of digestive enzyme activity, immune enzyme activity, and circadian rhythm of intestinal bacterial community structure in *E. carinicauda*. These findings may have important implications for optimizing feeding time and improving the quality of crustaceans in aquaculture environments. The results of this study provide a basis for further study on the physiological mechanism of diurnal variation of intestinal flora in crustaceans.

## Figures and Tables

**Figure 1 microorganisms-10-02258-f001:**
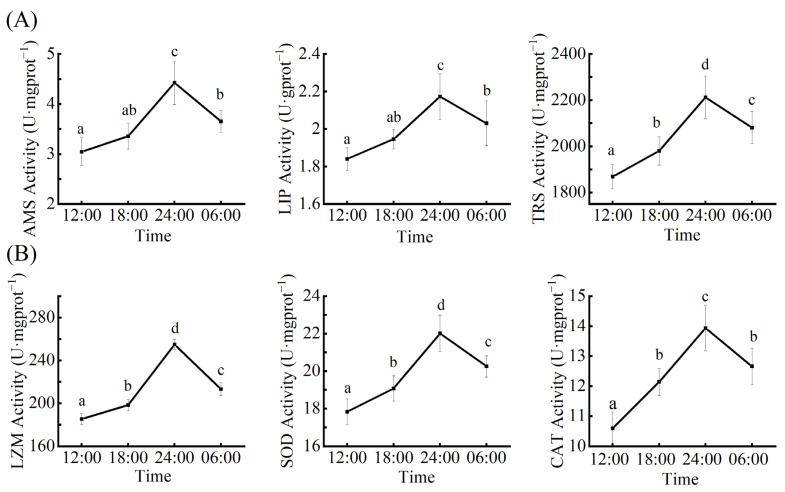
The Circadian Rhythm of enzyme activities. (**A**) α-amylase (AMS) activity diagram, Lipase (LIP) activity diagram, and Trypsin (TRS) activity diagram. (**B**) Lysozyme (LZM) activity diagram, Superoxide dismutase (SOD) activity diagram, and Catalase (CAT) activity diagram.

**Figure 2 microorganisms-10-02258-f002:**
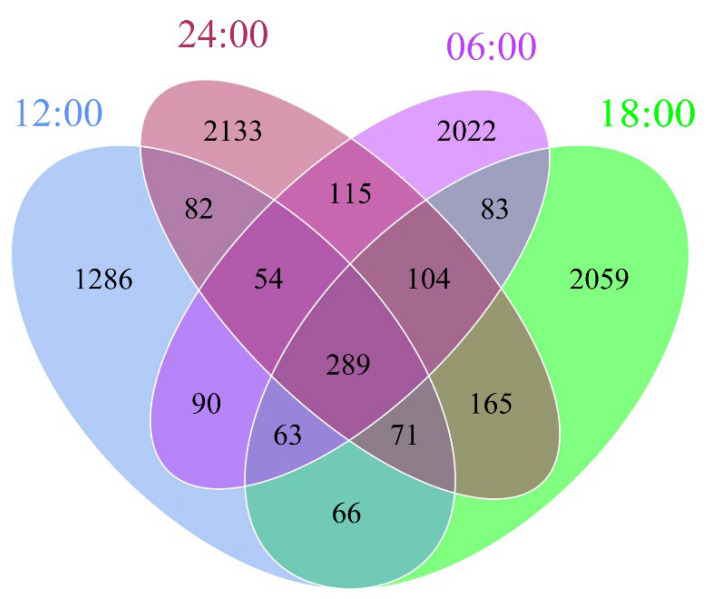
Venn diagram showing the number of unique and shared ASVs among the different sample times.

**Figure 3 microorganisms-10-02258-f003:**
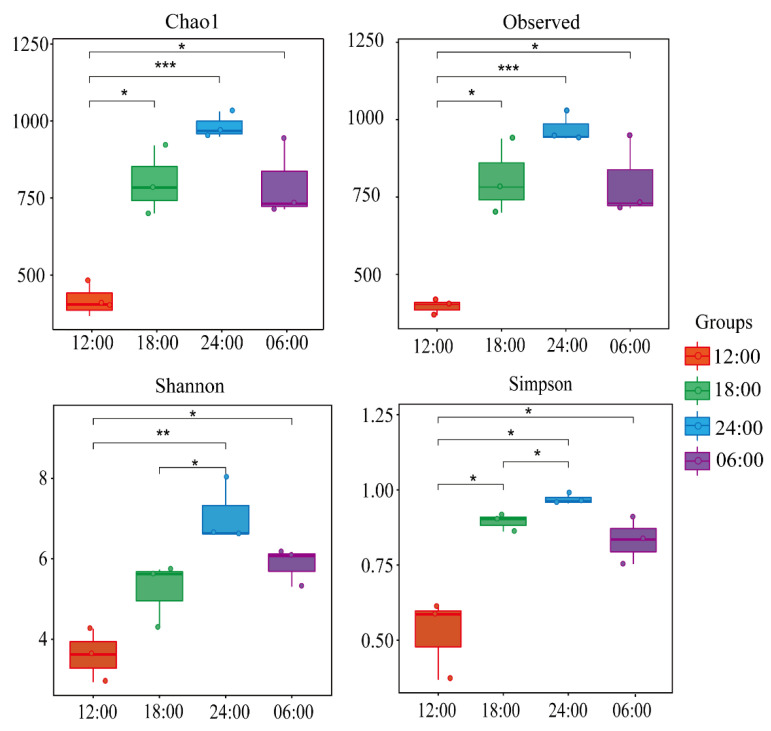
Alpha-diversity indices of the gut bacterial community (Chao1, Observed, Shannon, Simpson. confidence intervals (whiskers). Asterisks indicate statistically significant differences between pairs of values (* *p* < 0.05, ** *p* < 0.01, and *** *p* < 0.001).

**Figure 4 microorganisms-10-02258-f004:**
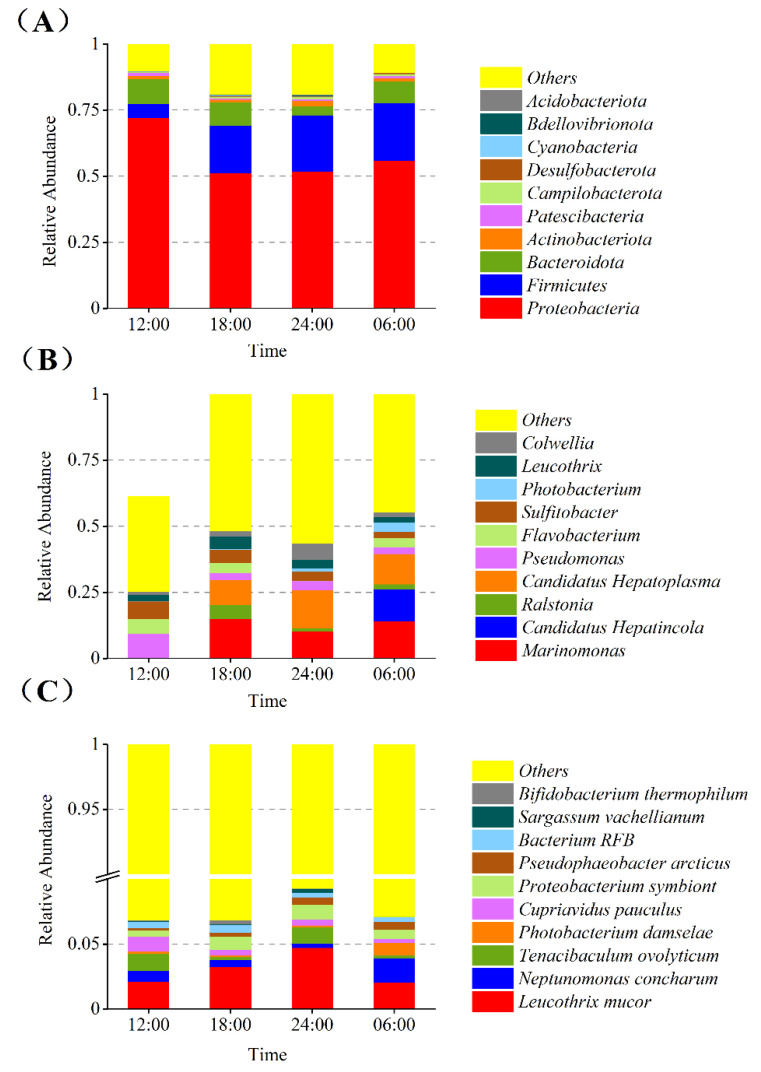
The relative abundance of the gut bacterial community. (**A**) Phylum levels. (**B**) Genus levels. (**C**) Species levels.

**Figure 5 microorganisms-10-02258-f005:**
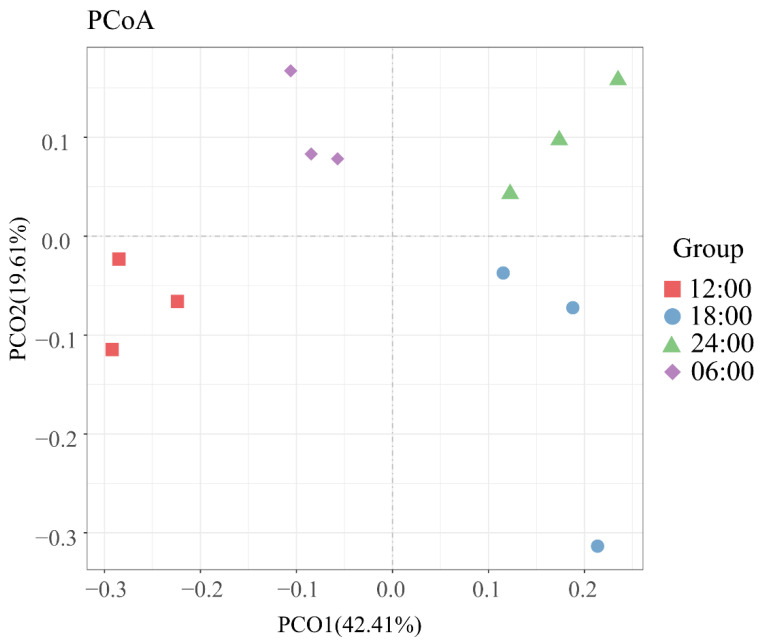
Principal coordinate analysis (PCoA) of the gut bacterial community.

**Figure 6 microorganisms-10-02258-f006:**
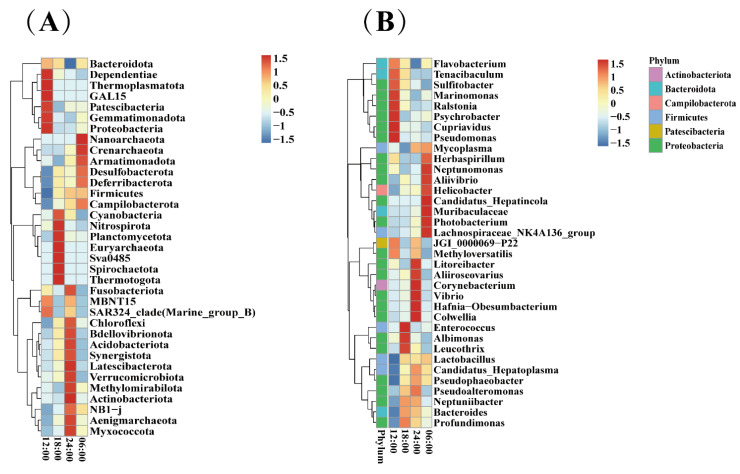
(**A**) Correlation analyses of the gut bacteria (phylum level). (**B**) Correlation analyses of the gut bacteria (genera level).

**Figure 7 microorganisms-10-02258-f007:**
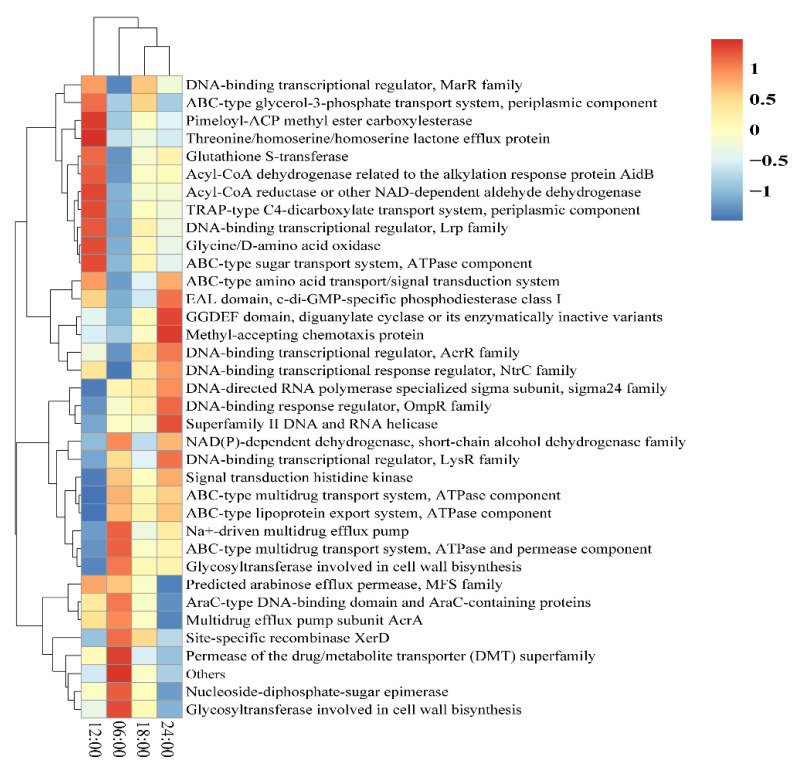
The relative abundance of gene function.

## Data Availability

Not applicable.

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
