# Peer review of "Comparative Analyses of the Exopalaemon carinicauda Gut Bacterial Community and Digestive and Immune Enzyme Activity during a 24-Hour Cycle"

_microorganisms, 2022, doi:10.3390/microorganisms10112258_

Round 1

Reviewer 1 Report

The authors examined levels of certain gut and tissue enzymes, and bacterial community composition, in as shrimp species at four different time points during the day. Substantial differences were noted, and these were ascribed to a circadian rhythm within the animal.

            The reviewer’s primary concern regarding the manuscript is the authors’ failure to convince the reviewer that the differences noted were due to the circadian rhythm, as opposed to other factors that were not considered (and not described in the methods). For example, the timing of feed presentation, or the light/dark cycle used in the cultivation conditions, were not described. Could these have had an impact, independent of (or perhaps in association with) the purported circadian rhythm? The authors need to address these potential confounding factors, by providing the missing information in the Methods section, and in the discussion of the data. A second concern regards confusion over the number of animals versus the number of samples (see comment to P3, Section 2.4 below). A third concern regards the weak effort to tie the microbiological, enzymatic, and gene expression data to any biological consequences. What is the importance of a particular taxon’s abundance being elevated at a particular time point? Do these levels have anything to do when the shrimp were fed versus when they were sampled? For the the PiCRust data, can anything really be concluded? The reviewer gets the feeling that the technique was used merely because it was available, rather than its having of any predictive or explanatory value.

Specific comments:

P2, para.2, L1: The first clause of this sentence makes no sense; please rewrite.

P3, Section 2.1: More detail is needed here. How were the shrimp synchronized to the 24 h cycle? What light/dark regimen was used? When were they fed relative to sampling?

Were the shrimp sacrificed at time of sampling (as opposed to withdrawal of intestinal contents or blood from live animals)? If the former, were the shrimp euthanized, and if so, what method was used?

P3, Section 2.2, L4-5: Describe the grinder (manufacturer, model) and grinding conditions (time, temperature, intensity). Express centrifugation conditions on an relative centrifugal force (RCF, x g), not as a rotor speed, as a given rotor speed will yield different RCF values depending on rotor diameter and geometry. What assay was used for protein, and what was the protein standard?

P3, Section 2.4: If there were 20 individuals per sample time, and 4 sample times, why were there 12 sequencing libraries?

P3, Section 2.5, L6: What version of SILVA was used?

P4-5, Sections 3.1 and 3.2: There’s a lot of verbiage here, considering the simple message in Figure 1 that all of the enzyme activities were at their minima at 12:00, and all peaked at 24:00. Not much more needs to be said than that, because the letters on the data points -- indicating significant differences among treatments -- fill in the details for the readers.

P8, Section 3.6, L7-11: The authors seem to be referring here to the greater dispersion of points within the group, but this dispersion is really just due to the single point at the lower right of the figure. What seems more important to this reviewer is that the all four time points show different community compositions from each other, but the authors do not even mention this.

P9, Section 3.7. and Figure 6: More explanation is needed for the Figure. What do the numerical values in the key represent? Are these LDA scores from a LEfSe analysis? In the text, what is meant by “high” abundance? And why are only some taxa from the figure mentioned, while others are not (for example, from Fig.6 at 6:00, Neptunimonas, Muribaculaceae, and Photobacterium are mentioned in the text, but Aliivibrio, Candidatus Hepatincola, but Lachnospiraceae NK4A136 are not)?

P11, Section 4, para.2, L4-7: Are these true circadian rhythms, or are they merely responses to a particular feeding schedule (that was not described in the methods)?

P12, para.2, L3: What, if anything, does this actually mean for the animal?

Figures 1 through 5: These figures could be improved by increasing the font size of the axis labels, and using a sans-serif font. Figure 4 is particularly difficult to read – the key listing the individual taxa should be expanded to make it more readable.

Minor edits:

P1, Abstract, L4-5: Provide the names of the enzymes, as opposed to abbreviations.

P1, para.1, L2: Change “is” to “are”.

P1, para.1, L13: Here and elsewhere, the species name should not be capitalized.

P2, para.3, L1: Delete “ridge trail”?

P2, pare.3, L4: Change “behaxior” to “behavior”.

P3. Section 2.2, L2: Suggest changing abbreviation for lipase from LPS to something else, perhaps LIP, as “LPS” is a common abbreviation for lipopolysaccharide, an important immunomodulator.

P7, Section 3.5, L8: Change “genus” to “genera”.

P8, Section 3.6, L1: Insert “represented by” after “is”.

P8, Figure 4 legend: Change “genera” to “genus”.

P10: Section 3.8, L2: Change “Stats” to “States”.

P12, para.1: Italicize all (not just some) genus and species names.

P12, para.1, L16: Should “Bifidobacteria” be “Bifidobacterium”?

Author Response

Dear Editor and reviewers,

Thank you for your letter and the reviewers’ comments concerning our manuscript entitled “Comparative analyses of the Exopalaemon carinicauda gut bacterial community and digestive and immune enzyme activity during a 24-hour cycle” (microorganisms-1980086). Those comments are valuable and very helpful for revising and improving our manuscript as well as important for guiding the significance of our research. We have read through the comments carefully and made corrections. Based on the instructions provided in your letter, we uploaded the file of the revised manuscript. The revised portions are indicated in red in the manuscript. The responses to the reviewer’s comments are as follows.

Point 1: The reviewer’s primary concern regarding the manuscript is the authors’ failure to convince the reviewer that the differences noted were due to the circadian rhythm, as opposed to other factors that were not considered (and not described in the methods). For example, the timing of feed presentation, or the light/dark cycle used in the cultivation conditions, were not described. Could these have had an impact, independent of (or perhaps in association with) the purported circadian rhythm? The authors need to address these potential confounding factors, by providing the missing information in the Methods section, and in the discussion of the data.

Response 1: We sincerely appreciate the valuable comments. To reduce the influence of other factors. All individuals were transported to the Jiangsu Key Laboratory of Marine Biotechnology of Jiangsu Ocean University and acclimated for two weeks in environmentally controlled tanks (salinity 25, temperature 25, natural light period). During acclimation, the prawns were temporarily fed twice daily (07:00, 18:00) with artificial diets until 24 h before use. The results showed that the digestive and immune enzyme activities of E. carinicauda changed significantly at four-time points throughout the day, with the highest value at 24:00 and the lowest value at 12:00. Similar results were observed in Eriocheir sinensis (Yu C, Li L, Jin J, et al. Comparative analysis of gut bacterial community composition during a single day cycle in Chinese mitten crab (Eriocheir sinensis)[J]. Aquaculture Reports, 2021, 21: 100907.). Sinonovacula constricta has shown circadian rhythmicity in digestive enzymes under light/dark cycle conditions (Liu Y, Yao H-h, Zhou T, et al. The Discovery of Circadian Rhythm of Feeding Time on Digestive Enzymes Activity and Their Gene Expression in Sinonovacula constricta Within a Light/Dark Cycle. Frontiers in Marine Science.2021, 8,744212.). Circadian rhythmicity is a defining feature of mammalian metabolism that synchronizes metabolic processes to day-night light cycles. Kuang et al showed that the intestinal microbiota programs diurnal metabolic rhythms in the mouse small intestine through histone deacetylase 3 (HDAC3) (Kuang Z, Wang Y, Li Y, et al. The intestinal microbiota programs diurnal rhythms in host metabolism through histone deacetylase 3[J]. Science, 2019, 365(6460): 1428-1434.). Some research showed that This circadian synchronization is fundamental to metabolic processes that must be coupled to diurnal sleep-wake and feeding-fasting cycles. We have supplemented this information in the materials and methods section (P2, Section 2.1) and discussion section (P11, para.2, L9-16 and P12, para.2, L18-21, para.3, L6-8). In future studies, we will consider exploring the relationship between more factors and the circadian rhythm of aquatic animals.

Point 2: A second concern regards confusion over the number of animals versus the number of samples (see comment to P3, Section 2.4 below).

Response 2: We really appreciate your suggestions. Since the gut and stomach of E. carinicauda are relatively small, we need to meet the requirements of enzyme activity assay and sequencing as much as possible. In this case, some samples may contain different amounts of intestinal and stomach tissue. In addition, not all three tissues of E. carinicauda were successfully collected at the time of sampling. We will increase the sampling level in future experiments.

Point 3: A third concern regards the weak effort to tie the microbiological, enzymatic, and gene expression data to any biological consequences.

Response 3: We really appreciate your suggestions. The change in life activities throughout a cycle of approximately 24 hours is called the circadian rhythm. Circadian rhythm has an important impact on biological metabolism, digestion, immunity, and other physiological activities. Our study showed that the digestive and immune enzyme activities of E. carinicauda changed significantly at four-time points throughout the day, with the highest value at 24:00 and the lowest value at 12:00. The structure, species diversity, relative abundance, and function of the gut microbiota were varied at different times of the day. However, the relationship between enzyme activity and bacterial flora still requires some kind of mediator, most likely metabolic substances. In the study of Sinonovacula constricta, the relative expression of digestive enzyme genes shared a similar pattern with the activities of digestive enzymes (Liu Y, Yao H, Zhou T, et al. The discovery of circadian rhythm of feeding time on digestive enzymes activity and their gene expression in Sinonovacula constricta within a light/dark cycle. [J]. Frontiers in Marine Science, 2021: 1478.). In future studies, we will try our best to tie the microbiological, enzymatic, and gene expression data.

Point 4: What is the importance of a particular taxon’s abundance being elevated at a particular time point? Do these levels have anything to do when the shrimp were fed versus when they were sampled?

Response 4: Thanks for your friendly reminder. Intestinal microorganisms in an organism are interdependent and mutually restricted, and maintain the homeostasis of the internal environment. As an indivisible ‘organ’ of the organism, the gut bacterial community has a great influence on the physiological activity of the host, such as immunity and growth, and its composition is related to many factors such as heredity, nutrition, and the environment. The abundance of particular taxon should be to meet the body's needs, such as digestion, immunity, metabolism, and so on. These microbiotas have the potential to be developed into probiotics. However, it is still difficult to locate a specific species, and that is what we are trying to do. This study that the intestinal bacterial community structure was different at different time points. The most abundant genera are Marinomonas, Ralstonia, Candidatus hepatoplasma and Candidatus hepatincola. Bifidobacterium thermophilus is an important intestinal beneficial bacteria, which plays an important role in nutrition, immune enhancement and improving gastrointestinal function. All individuals were acclimated for two weeks in environmentally controlled tanks (salinity 25, temperature 25, natural light period). During acclimation, the prawns were temporarily fed twice daily (07:00, 18:00) with artificial diets until 24 h before use. In future studies, we will consider exploring the effects of feeding and sampling on the abundance of the microbiota.

Point 5: For the the PiCRust data, can anything really be concluded? The reviewer gets the feeling that the technique was used merely because it was available, rather than its having of any predictive or explanatory value.

Response 5: We really appreciate your professional comments. PICRUSt2 integrates existing open-source tools to predict genomes of environmentally sampled 16S rRNA gene sequences (Douglas G M, Maffei V J, Zaneveld J, et al. PICRUSt2: An improved and customizable approach for metagenome inference[J]. BioRxiv, 2020: 672295.). ASVs are placed into a reference tree, which is used as the basis of functional predictions. The PICRUSt2 default genome database is based on 41,926 bacterial and archaeal genomes from the Integrated Microbial Genomes (IMG) database (Douglas G M, Maffei V J, Zaneveld J R, et al. PICRUSt2 for prediction of metagenome functions[J]. Nature Biotechnology, 2020, 38(6): 685-688.). Based on the ASV tree and ASV gene information in the Greengene database, the analysis method inferred the gene function spectrum of their common ancestor. At the same time, the gene function spectrum of other unknown species in the Greengene database was inferred, and the gene function prediction spectrum of the whole lineage of archaea and bacteria domain was constructed. Finally, the microbiota composition obtained by sequencing is "mapped" to the database, so as to predict the metabolic function of the microbiota. Although the software did not perform well in annotating the function of the microbiota, better analysis tools have not been developed. Perhaps it would be more helpful to screen for a single strain.

Specific comments:

Point 1: P2, para.2, L1: The first clause of this sentence makes no sense; please rewrite.

Response 1: We really appreciate your suggestions. This sentence has been deleted.

Point 2: P3, Section 2.1: More detail is needed here. How were the shrimp synchronized to the 24 h cycle? What light/dark regimen was used? When were they fed relative to sampling?

Response 2: We sincerely appreciate the valuable comments. Before the experiment, the light conditions were not intervened in the adaptation stage (two weeks) of the laboratory and during the experiment, and the light of the breeding environment changed with natural time. During acclimation, the shrimp were fed twice daily with commercial shrimp food until 24 h before use. We have supplemented this information in the materials and methods section (P2, Section 2.1 L7-9).

Point 3: Were the shrimp sacrificed at time of sampling (as opposed to withdrawal of intestinal contents or blood from live animals)? If the former, were the shrimp euthanized, and if so, what method was used?

Response 3: We really appreciate your suggestions. The experimental protocols and procedures for animal husbandry and handling were performed according to the Animal Care Committee of Jiangsu Ocean University in the present study. The shrimp were euthanized after anesthesia by the anesthetic (alcohol: eugenol = 10:1) and the intestinal tract, hepatopancreas, and gastric tissues of individuals were sampled and immediately preserved in liquid nitrogen. We have supplemented this information in the materials and methods section (P2, Section 2.1 L1-3, 10-11).

Point 4: P3, Section 2.2, L4-5: Describe the grinder (manufacturer, model) and grinding conditions (time, temperature, intensity). Express centrifugation conditions on an relative centrifugal force (RCF, x g), not as a rotor speed, as a given rotor speed will yield different RCF values depending on rotor diameter and geometry. What assay was used for protein, and what was the protein standard?

Response 4: Thanks for your friendly reminder. Weighed samples were mixed with 9 times its volume of normal saline (0.9%), which was homogenized for 20 seconds at 4 ° C and 20 Hz/s using an automatic grinding machine (Tissuelyser-24, China). Centrifuge the sample at 4 ° C and 1500×g for 10 minutes using a refrigerated centrifuge (MIKRO 220R, Hettich). Enzyme extracts were kept at −80 °C until they were analyzed within 48 hours of extraction. Total protein content and all enzymatic activity analyses were conducted using commercial assay kits (Jiancheng Biological Engineering Institute, Nanjing, China). The total protein content was determined by Coomassie brilliant blue method. Total protein content and all enzymatic activity analyses were conducted using commercial assay kits (Jiancheng Biological Engineering Institute, Nanjing, China). We have supplemented this information in the materials and methods section (P2, Section 2.2 L3-12).

Point 5: P3, Section 2.4: If there were 20 individuals per sample time, and 4 sample times, why were there 12 sequencing libraries?

Response 5: We really appreciate your suggestions. Since the gut and stomach of E. carinicauda are relatively small, we need to meet the requirements of enzyme activity assay and sequencing as much as possible. In this case, some samples may contain different amounts of intestinal and stomach tissue. In addition, not all three tissues of E. carinicauda were successfully collected at the time of sampling. We will increase the sampling level in future experiments.

Point 6: P3, Section 2.5, L6: What version of SILVA was used?

Response 6: Thanks for your friendly reminder. We have supplemented this information in the revised manuscript (P3, Section 2.5, L 6-7).

Point 7: P4-5, Sections 3.1 and 3.2: There’s a lot of verbiage here, considering the simple message in Figure 1 that all of the enzyme activities were at their minima at 12:00, and all peaked at 24:00. Not much more needs to be said than that, because the letters on the data points --indicating significant differences among treatments -- fill in the details for the readers.

Response 7: We sincerely appreciate the valuable comments. We have made changes in the revised manuscript (P4-5, Sections 3.1 and 3.2).

Point 8: P8, Section 3.6, L7-11: The authors seem to be referring here to the greater dispersion of points within the group, but this dispersion is really just due to the single point at the lower right of the figure. What seems more important to this reviewer is that all four-time points show different community compositions from each other, but the authors do not even mention this.

Response 8: We really appreciate your suggestions. Using a series of eigenvalues and eigenvectors, principal coordinate analysis (PCoA) extracts the most significant elements and structures from multidimensional data. An unweighted and weighted UniFrac-based PCoA of the cohort was performed to visually explore the similarity and variation between the samples’ microbial composition. As shown in figure 5, the PCoA results obviously showed that the bacterial communities of the samples from four-time points had significant differences. On the contrary, the diversity of bacterial communities within the group was low. We have supplemented this information in the materials and methods section (P8, Section 3.6 L6-9).

Point 9: P9, Section 3.7. and Figure 6: More explanation is needed for the Figure. What do the numerical values in the key represent? Are these LDA scores from a LEfSe analysis? In the text, what is meant by “high” abundance? And why are only some taxa from the figure mentioned, while others are not (for example, from Fig.6 at 6:00, Neptunimonas, Muribaculaceae, and Photobacterium are mentioned in the text, but Aliivibrio, Candidatus Hepatincola, but Lachnospiraceae NK4A136 are not)?

Response 9: We sincerely appreciate the valuable comments. The heat map of the correlation matrix comparing the time points, in terms of phylum-level abundance, further supported this daily change in gut bacterial community structure. The corresponding value of the heatmap is the Z value obtained after normalization of the relative abundance of species in each row. The Z value of a sample in a certain category is the difference between the relative abundance of samples in that category and the average relative abundance of all samples in that category divided by the standard deviation of all samples in that category. Multiple sequence alignment was performed using QIIME2 software. The absolute abundance of ASVs was normalized using a standard sequence number corresponding to the sample with the fewest sequences. According to the species annotation and abundance information of all samples at the genus level, Fig. 6 selects the top 35 genera of abundance. In the manuscript, we didn't list all the special categories of four time points. The results indicated that both the phylum- and genus-level abundance of the bacterial communities varied significantly during the daily cycle. We have supplemented this information in the materials and methods section (P9, Section 3.7 L5-6).

Point 10: P11, Section 4, para.2, L4-7: Are these true circadian rhythms, or are they merely responses to a particular feeding schedule (that was not described in the methods)?

Response 10: We really appreciate your suggestions. During acclimation, the prawns were temporarily fed twice daily (07:00, 18:00) with artificial diets until 24 h before use. The results showed that the digestive and immune enzyme activities of E. carinicauda changed significantly at four-time points throughout the day, with the highest value at 24:00 and the lowest value at 12:00. Similar results were observed in Eriocheir sinensis (Yu C, Li L, Jin J, et al. Comparative analysis of gut bacterial community composition during a single day cycle in Chinese mitten crab (Eriocheir sinensis)[J]. Aquaculture Reports, 2021, 21: 100907.). We have supplemented this information in the materials and methods section (P2, Section 2.1 L7-9).

Point 11: P12, para.2, L3: What, if anything, does this actually mean for the animal?

Response 11: We really appreciate your suggestions. Circadian rhythmicity is a defining feature of mammalian metabolism that synchronizes metabolic processes to day-night light cycles. Kuang et al showed that the intestinal microbiota programs diurnal metabolic rhythms in the mouse small intestine through histone deacetylase 3 (HDAC3) (Kuang Z, Wang Y, Li Y, et al. The intestinal microbiota programs diurnal rhythms in host metabolism through histone deacetylase 3[J]. Science, 2019, 365(6460): 1428-1434.). The microbiota induced expression of intestinal epithelial HDAC3, which was recruited rhythmically to chromatin, and produced synchronized diurnal oscillations in histone acetylation, metabolic gene expression, and nutrient uptake. Some research showed that This circadian synchronization is fundamental to metabolic processes that must be coupled to diurnal sleep-wake and feeding-fasting cycles. For aquatic animals, it will play an important role in optimizing feeding times and improving digestion and nutrient utilization. The results of this study provide a basis for further study on the physiological mechanism of diurnal variation of intestinal flora in crustaceans.

Point 12: Figures 1 through 5: These figures could be improved by increasing the font size of the axis labels, and using a sans-serif font. Figure 4 is particularly difficult to read – the key listing the individual taxa should be expanded to make it more readable.

Response 12: Thanks for your friendly reminder. We have made changes in the revised manuscript.

Minor edits:

Point 13: P1, Abstract, L4-5: Provide the names of the enzymes, as opposed to abbreviations.

Response 13: We have made changes in the revised manuscript (P1, Abstract, L4-5).

Point 14: P1, para.1, L2: Change “is” to “are”.

Response 14: We have made changes in the revised manuscript (P1, para.1, L3).

Point 15: P1, para.1, L13: Here and elsewhere, the species name should not be capitalized.

Response 15: Thanks for your friendly reminder. We have made changes in the revised manuscript (P1, para.1, L3).

Point 16: P2, para.3, L1: Delete “ridge trail”?

Response 16: Thanks for your friendly reminder. We have made changes in the revised manuscript (P2, para.3, L1).

Point 17: P2, pare.3, L4: Change “behaxior” to “behavior”.

Response 17: We are very sorry for the incorrect writing and have made changes in the revised manuscript (P2, para.3, L4).

Point 18: P3. Section 2.2, L2: Suggest changing abbreviation for lipase from LPS to something else, perhaps LIP, as “LPS” is a common abbreviation for lipopolysaccharide, an important immunomodulator.

Response 18: Thanks for your friendly reminder. We have made changes in the revised manuscript (P3. Section 2.2, L2).

Point 19: P7, Section 3.5, L8: Change “genus” to “genera”.

Response 19: We have made changes in the revised manuscript (P7, Section 3.5, L8).

Point 20: P8, Section 3.6, L1: Insert “represented by” after “is”.

Response 20: We have made changes in the revised manuscript (P8, Section 3.6, L1).

Point 21: P8, Figure 4 legend: Change “genera” to “genus”.

Response 21: We have made changes in the revised manuscript.

Point 22: P10: Section 3.8, L2: Change “Stats” to “States”.

Response 22: We are very sorry for the incorrect writing and have made changes in the revised manuscript (P10: Section 3.8, L2).

Point 23: P12, para.1: Italicize all (not just some) genus and species names.

Response 23: We have made changes in the revised manuscript (P12, para.1).

Point 24: P12, para.1, L16: Should “Bifidobacteria” be “Bifidobacterium”?

Response 24: Thanks for your friendly reminder. We have made changes in the revised manuscript (P12, para.1, L16).

Reviewer 2 Report

In this manuscript Xie et al. report on activity of digestive enzymes (AMS, LPS, TRS) and immune enzymes (SOD, LZM, CAT), of the intestinal bacterial community of Exopalaemon carinicauda, highlighting significant changes related to the circadian rhythm of this crustacean. A topic still to be explored is addressed, which will be useful for improving the welfare of these animals and the aquaculture technology.

Overall, the approach taken is scientifically sound. The studies were performed at the proper methodological level, the results are substantiated and well discussed with a logical conclusion.

I only noticed a few small flaws in the exhibition form. Below, there are some suggestions that are aimed at improving the quality of the submitted manuscript.

Pag. 3: I suggest specifying the type of tissue samples used for the extraction of enzymes and DNA, respectively in sections 2.2 and 2.3

Fig.1: it would be clearer if divided into two parts (A for digestive enzymes and B for immune enzymes). Useful to indicate the acronyms also in the caption of the figure.

Fig.3 replace “Scheme 95” with “Simpson” and “Observed species” with “Observed” in the caption of the figure.

Furthermore:

I recommend that you carefully review the acronyms and related extended forms, standardizing their use throughout the text for a clearer reading.

I recommend reviewing the punctuation and the use of capital letters especially after the point.

Author Response

Dear Editor and reviewers,

Thank you for your letter and the reviewers’ comments concerning our manuscript entitled “Comparative analyses of the Exopalaemon carinicauda gut bacterial community and digestive and immune enzyme activity during a 24-hour cycle” (microorganisms-1980086). Those comments are valuable and very helpful for revising and improving our manuscript as well as important for guiding the significance of our research. We have read through the comments carefully and made corrections. Based on the instructions provided in your letter, we uploaded the file of the revised manuscript. The revised portions are indicated in red in the manuscript. The responses to the reviewer’s comments are as follows.

Point 1: Pag. 3: I suggest specifying the type of tissue samples used for the extraction of enzymes and DNA, respectively in sections 2.2 and 2.3

Response 1: We really appreciate your suggestions. The stomach was used to determine the activity of three digestive enzymes, and the hepatopancreas was used to determine the activity of three immune enzymes. We have made changes in the revised manuscript (P3, Section 2.2, L3-4).

Point 2: Fig.1: it would be clearer if divided into two parts (A for digestive enzymes and B for immune enzymes). Useful to indicate the acronyms also in the caption of the figure.

Response 2: We sincerely appreciate the valuable comments. We have recreated figure 1 in the revised manuscript.

Point 3: Fig.3 replace “Scheme 95” with “Simpson” and “Observed species” with “Observed” in the caption of the figure.

Response 3: Thanks for your friendly reminder. We have made changes in the revised manuscript (P7, Fig 3).

Point 4: Furthermore: I recommend that you carefully review the acronyms and related extended forms, standardizing their use throughout the text for a clearer reading. I recommend reviewing the punctuation and the use of capital letters especially after the point.

Response 4: We really appreciate your suggestions. We have read the manuscript carefully and corrected some irregularities in the revised manuscript.

Round 2

Reviewer 1 Report

The authors have made changes to their manuscript, but the main original points raised by the reviewer were not adequately addressed. Without this information, the reviewer cannot further evaluate the revised manuscript. Specifically:

1) When and how, exactly, were the shrimp fed? The authors state that in they were fed an artificial diet at 0700 and 1800 during an acclimation period. But what were they fed, and when, after the acclimation period, during the actual experiment? In the introduction the authors state that this species exhibits “nocturnal hunting behavior”. Was there any “hunting” involved during the experimental period? Was the experimental period only the 24 hours during which shrimp were collected for analysis? During this period, were they even fed at all?

2) The number of animals sampled is still unclear to the reviewer.  The reviewer’s original comment, and the authors’ response, is as follows:

“P3, Section 2.4: If there were 20 individuals per sample time, and 4 sample times, why were there 12 sequencing libraries? Response 5: We really appreciate your suggestions. Since the gut and stomach of E. carinicauda are relatively small, we need to meet the requirements of enzyme activity assay and sequencing as much as possible. In this case, some samples may contain different amounts of intestinal and stomach tissue. In addition, not all three tissues of E. carinicauda were successfully collected at the time of sampling. We will increase the sampling level in future experiments.”

This response is confusing to the reviewer. Are the authors saying that multiple shrimp were collected and their stomach contents (or intestinal contents) were pooled, so that each sample analyzed came from multiple animals? If so, how many shrimp were pooled per sample, and how many pooled samples were analyzed at each time point?

The reviewer looks forward to clarifications from the author, which should be included within the text of the manuscript (not just included in an authors' response)..

Author Response

Dear Editor and reviewers,

Thank you for your letter and the reviewers’ comments concerning our manuscript entitled “Comparative analyses of the Exopalaemon carinicauda gut bacterial community and digestive and immune enzyme activity during a 24-hour cycle” (microorganisms-1980086). Those comments are valuable and very helpful for revising and improving our manuscript as well as important for guiding the significance of our research. We have read through the comments carefully and made corrections. Based on the instructions provided in your letter, we uploaded the file of the revised manuscript. The revised portions are indicated in green in the manuscript. The responses to the reviewer’s comments are as follows.

Point 1: When and how, exactly, were the shrimp fed? The authors state that in they were fed an artificial diet at 0700 and 1800 during an acclimation period. But what were they fed, and when, after the acclimation period, during the actual experiment? In the introduction the authors state that this species exhibits “nocturnal hunting behavior”. Was there any “hunting” involved during the experimental period? Was the experimental period only the 24 hours during which shrimp were collected for analysis? During this period, were they even fed at all?

Response 1: We sincerely appreciate the valuable comments. During the two-week acclimation phase, the shrimp were temporarily fed twice daily (07:00, 18:00) with artificial compound feed (Haida, China) at a daily feeding rate of 5% body weight. Feeding was stopped 24 h before the experiment. During the experimental period, 20 individuals were randomly collected at four time points, at intervals of 6 h (12: 00, 18: 00, 24: 00, and 6: 00). It is well known that many crustaceans are nocturnal predators, which is a manifestation of the circadian rhythm. As an economically important shrimp breeding species, E. carinicauda exhibits nocturnal hunting behavior. The results showed that the digestive and immune enzyme activities of E. carinicauda changed significantly at four-time points throughout the day, with the highest value at 24:00 and the lowest value at 12:00. These results help to reveal the nocturnal predation habits of crustaceans. Of course, the study of shrimp predation behavior is also very interesting. We have supplemented this information in the materials and methods section (P2, Section 2.1 L7-10). In future studies, we will consider sampling shrimp during normal feeding and observing the predation behavior of shrimp.

Point 2: The number of animals sampled is still unclear to the reviewer.  The reviewer’s original comment, and the authors’ response, is as follows:

“P3, Section 2.4: If there were 20 individuals per sample time, and 4 sample times, why were there 12 sequencing libraries? Response 5: We really appreciate your suggestions. Since the gut and stomach of E. carinicauda are relatively small, we need to meet the requirements of enzyme activity assay and sequencing as much as possible. In this case, some samples may contain different amounts of intestinal and stomach tissue. In addition, not all three tissues of E. carinicauda were successfully collected at the time of sampling. We will increase the sampling level in future experiments.”

This response is confusing to the reviewer. Are the authors saying that multiple shrimp were collected and their stomach contents (or intestinal contents) were pooled, so that each sample analyzed came from multiple animals? If so, how many shrimp were pooled per sample, and how many pooled samples were analyzed at each time point?

The reviewer looks forward to clarifications from the author, which should be included within the text of the manuscript (not just included in an authors' response).

Response 2: We really appreciate your professional comments. During the experimental period, 20 individuals were randomly collected at four time points, at intervals of 6 h (12: 00, 18: 00, 24: 00, and 6: 00). As we all know, each experimental assay may require different sample volumes. For only 1 gram of E. carinicauda, the sample size of one shrimp can not meet the requirements of enzyme activity or intestinal flora detection. In this case, we have to mix multiple samples from the same group. The gut tissue of the fasting shrimp is nearly transparent, which is not easy to obtain smoothly. Twenty individuals were randomly collected at four time points. We successfully obtained intestinal tissue from 18 individuals at each sampling site, which was evenly divided into three groups. We have supplemented this information in the materials and methods section (P3, Section 2.2 L3-5, Section 2.4 L1-4). This sampling method is actually very common in the study of crustaceans, not only to meet the detection needs but also to help reduce the impact of individual differences on the results.

Thank you again for your positive comments and valuable suggestions to improve the quality of our manuscript.

Sincerely yours

Dr. Panpan Wang

E-Mail: [email protected]

Jiangsu Ocean University
